

# Generalized relational tensors for chaotic time series

Vasilii A. Gromov, Yury N. Beschastnov and Korney K. Tomashchuk

School of Data Analysis and Artificial Intelligence, Higher School Economics University, Moscow, Russia

## ABSTRACT

The article deals with a generalized relational tensor, a novel discrete structure to store information about a time series, and algorithms (1) to fill the structure, (2) to generate a time series from the structure, and (3) to predict a time series. The algorithms combine the concept of generalized z-vectors with ant colony optimization techniques. To estimate the quality of the storing/re-generating procedure, a difference between the characteristics of the initial and regenerated time series is used. For chaotic time series, a difference between characteristics of the initial time series (the largest Lyapunov exponent, the auto-correlation function) and those of the time series re-generated from a structure is used to assess the effectiveness of the algorithms in question. The approach has shown fairly good results for periodic and benchmark chaotic time series and satisfactory results for real-world chaotic data.

# INTRODUCTION

Discrete models used to represent chaotic time series make it possible to study the time series with the employment of complex networks theory, thus allowing deeper insight into time series dynamics.[1] Here and after, the term "discrete models" ("discrete time-series models") is used to denote set of models used to represent time series: chaotic graphs, power spectrum, visibility graph, recurrence matrix, ordinal networks, recurrence networks, visibility graphs, transition networks, *etc.*

The central idea of this work is the assertion that the appropriate way to assess the quality of the representation of a time series (no matter how this representation is formed) is to measure quantitative difference (see Eq. (1)) between the characteristics of the original and restored time series. Here and after, the term "characteristics" denote the set of metrics which store information about the structure of the time series under the study. These characteristics may include: Lyapunov spectrum, dimensions of attractors, entropy-complexity, partial autocorrelation function, *etc.* If the quantitative difference between Lyapunov spectrum, attractor dimensions, *etc.* of the original time series and the synthetic time series generated by the discrete model is not met, then any judgments about the time series, made based on the model with which the time series is represented (graph, tensor, array, *etc.*), are groundless. In the literature known to us, such problem statement has not been encountered, thus we believe it is right to present it to the readers.

Corresponding authors
Vasilii A. Gromov,
stroller@rambler.ru
Korney K. Tomashchuk, korney-tomashchuk@yandex.ru

[1] This class of models compares favorably with a neural networks models (Transformer and Time2Vec, for example) and many other AI models in interpretability and explainability: such models make it possible to examine the discrete time-series models and draw conclusions about type and structure of the dynamical process under study.

[2] We mean that when generating a series from a discrete model, the expert knowledge of the user of the algorithm, *etc.*, is not used, which happens in some approaches.

[3] In this case, a sliding window used to obtain z-vectors is replaced by a set of sliding combs with some teeth broken off (patterns). One should stress here that the Takens's theorem defines the length of z-vector, not the way one embeds a time series into a space of generalized z-vectors (*Small, 2005*). On the other hand, *Gromov & Borisenko (2015)* show that the generalized z-vectors may be more efficient for regularly sampled time series.

At the same time, we believe that only the information encoded in the model should be used to restore the series. The use of other information[2] is incorrect from the point of view of the problem statement. It seems to us, perhaps somewhat presumptuous, that such validation (evaluation of the quantitative difference of the characteristics of the initial and reconstructed time series) should become the standard when working with any representation of the time series (the choice of characteristics considered, as it seems to us, is a matter of the future and requires the joint efforts of the entire community).

The second goal that we pursued in this article is to show that solving the problem in such a formulation is possible: we have presented a new type of model and algorithms for filling/restoring it (the prediction algorithm is a side effect here) that meets the specified requirements. A generalized relation tensor relies on the concept of a generalized z-vector, a z-vector that comprises non-consecutive observations of a time series (irregular embedding). The tensor is readily applicable to both regularly and irregularly sampled time series (with missing data). To design fast and efficient algorithms to fill, re-generate, and predict with the tensor we utilize ant colony optimization (distributed artificial intelligence approach). We evaluated the quality of the resulting tensors and the corresponding algorithms by computing quantitative difference between chosen characteristics of the original and reconstructed time series.

The idea to represent a time series using some discrete model (which is of particular value for chaotic time series) has made great progress over the past few decades (*Budroni, Tiezzi & Rustici, 2010*; *Kulp, 2013*; *Lacasa et al., 2008*; *Marwan et al., 2009*; *McCullough et al., 2017*). We should emphasize that regardless of the model or method used any employed approach can be considered as a way of representing a time series with the use of a relatively small number of real numbers—time series analysis is essentially data compression (*Bradley & Kantz, 2015*; *Zou et al., 2019*).

In addition to the objective function formulated above for assessing the quality of the tensor, we must formulate some constraints. The accumulated experience of using such discrete models has revealed several requirements. Firstly, it is sometimes necessary to restrict the number of vertices and edges. Such restriction excludes the approaches where the number of vertices equals to the number of observations. The latter is particularly the case for chaotic systems, where it is supposed that the time series is long enough to attain regions of strange attractor with relatively small invariant measure values (*Kantz & Schreiber, 2003*; *Malinetskii & Potapov, 2000*). This brought into existence several methods that rely on the concept of z-vector (regular embedding); please refer to *Mutua, Gu & Yang (2016)* and *Laut & Räth (2016)*. A z-vector is a sequence of consecutive observations of a time series (a segment of a time series). Its length is bounded from below by the minimum embedding dimension required to produce a delay embedding such that topological equivalence is guaranteed (according to Takens's theorem); in actual practice, one should apply sophisticated methods to estimate this length. Secondly, such an algorithm is practicable only if it can process irregularly sampled data (*Amigó, Kennel & Kocarev, 2005*). This implies that one should modify the concept of irregular embedding (*Small, 2005*) by employing generalized z-vectors, composed of non-consecutive observations.[3] Finally, the third requirement is that such model must be able to generate a time series. This makes

it possible to compare characteristics of original and re-generated time series to estimate, whether the model is consistent with the corresponding time series (*McCullough et al., 2017*). The relation tensor proposed in the article satisfies the indicated constraints and allows optimizing the objective function.

The novel models compares favorably with analogous discrete models used to represent a time series:

1. The tensor generates a time series with approximately the same 'chaotic' characteristics as the original one without resorting to observations of the original time series. Hence one may be sure that one analyses a discrete model associated with this very time series, not with other chaotic time series.

2. The number of elements in the tensor is determined by the number of possible generalized z-vectors (see below), and hence it is rather large. To avoid combinatorial explosion, we propose to use algorithms based on the ant colony optimization method; this makes it possible to develop efficient algorithms to re-generate a time series from the tensors and predict it.

3. The model allows one to deal with both regularly and irregularly sampled data.

The rest of the article is organized as follows. The next section reviews models utilized to represent information about a time series and methods to fill them. The third section states the problem; the fourth one introduces the concept of a generalized relational tensor. The methods employed are discussed in greater detail in the fifth section. The sixth section provides results for noisy periodic time series, standard chaotic time series, and real-world chaotic data. Finally, the last section presents conclusions.

## LITERATURE REVIEW

*Zou et al. (2019)* review various discrete models used to represent a non-Markovian time series: recurrence networks, visibility graphs, and transition networks. Recurrence networks (*Marwan et al., 2009*; *Zou et al., 2019*) employ recurrent methods to fill graphs. The popular model used to represent information about a time series is a visibility graph (*Lacasa et al., 2008*; *Luque et al., 2009*). *Flanagan & Lacasa (2016)* consider visibility graphs associated with financial time series as complex networks. The Kullback–Leibler divergence calculated for distribution functions associated with a horizontal visibility graph (HVG) and a visibility graph allows estimation of the irreversibility of the respective time series. *Gonçalves et al. (2016)* propose a way to extract information from HVG, based on amplitude differences, and show that this approach results in better characterization of real-world dynamical systems (El-Nino Southern Oscillation), making it possible to represent properly features important for climatologists. In addition, it requires a significantly shorter time series.

*Zhuang, Small & Feng (2014)* also deal with visibility graphs as complex networks. The authors address the problem of community detection for these graphs in order to reveal some properties of the time series. *Lan et al. (2015)* propose a fast method to build a visibility graph for a given time series. *Li et al. (2016)* construct visibility graphs for the time series of fractional Brownian motions, while *Gonçalves et al. (2016)* estimate information measures for the same time series with the employment of its visibility graph. *Bezsudnov & Snarskii*

*(2014)* propose a parametric visibility graph (the generalization of a conventional one) and examine the influence of the view angle (the parameter) on distributions associated with the graph. Extension that can cope with non-stationary time series is discussed in the article by *Gao et al. (2017)*. *Budroni, Tiezzi & Rustici (2010)* utilize a specific graph, named a chaotic graph, to describe a time series.

Transition networks algorithms employ the transition matrix between various elements of a time series (observations, motifs, *etc.*) in order to describe the system dynamics. For instance, *McCullough et al. (2017)* construct the ordinal networks with the employment of the Bandt and Pompe *Rosso et al. (2007a)* representation for chaotic time series. The algorithms for compression and re-generation are also presented; in order to assess the quality of the compression characteristics of chaotic time series such as the largest Lyapunov exponent, correlation integral, *etc.* were used. Unfortunately, the regeneration algorithm employs observations of the original time series. *Sakellariou, Stemler & Small (2019)* discuss algorithms to construct an ordinal network and to explore model thereof (associated with regular and chaotic time series) as complex networks; this allows finding some intriguing relations between time series characteristic and ordinal network features. *Keller & Sinn (2009)* employ ordinal patterns in order to design an efficient way to estimate dynamical systems complexity (with the employment Kolmogorov–Sinai entropy). *Amigó, Kennel & Kocarev (2005)* prove equivalency of metric and permutation entropy rates discrete-time stationary stochastic processes that make it possible to use them later as an estimate for the former.

*Nicolis, Cantu & Nicolis (2005)* discuss non-Markovian transition networks used for chaotic time series representation. This approach can be easily extended to work with irregularly sampled data (*Kulp et al., 2016*; *McCullough et al., 2016*; *Sakellariou et al., 2016*). *Kulp (2013)* modifies a method originally proposed by *Wiebe & Virgin (2012)*; the method employs a time series spectrum. *Gromov & Shulga (2012)* utilize multigraph to represent information about time series. *Martinez Alvarez et al. (2011)* employ (unique) adjacent matrices of z-vectors (graphlets) as nodes, while the edges are established if the second z-vector follows the first. *Laut & Räth (2016)* construct a complex network in order to test whether or not a time series is nonlinear.

It is worth noting that one may relate the problem to represent a chaotic time series using the generalized relation tensors, proposed in this article, with a popular problem to design recommendation system. In particular, it is germane to compare with (decentralized and collaborative) clustering bandits algorithms (*Li, 2016*). Indeed, information that a single ant receives during one step (in the present article) corresponds to information that a single agent receives at one time round (in clustering bandits); a tensor element that is filled by various ants at various moments correspond to a cluster of bandits and so on. *Korda, Szorenyi & Li (2016)* discuss distributed confidence ball algorithms for solving linear bandit problems in peer-to-peer networks. Authors assume that all bandits solve the same problem; this seems to limit the utility of the article for our purpose. *Hao et al. (2015)*, to solve a similar problem, employs clustering to group users in *ad-hoc* social network environments. *Mahadik et al. (2020)* deal with a scalable algorithm for the problem, DistCLUB. *Gentile, Li & Zappella (2014)* investigate online clustering; the article presents a strict analysis of

the problem. *Li, Karatzoglou & Gentile (2016)* examine collaborative effects. One should emphasize that, for all the above articles, a payoff received by an agent is determined by a linear function (usually, noised). Whereas, for the present article, a payoff is nonlinear and determined by a time series in question. This fact interferes with the direct comparison of the algorithm for the two problems.

The idea of representing time series with a tensor is not new. This technique is a common tool in machine learning. Such that in the article (*Yang, Krompass & Tresp, 2017*) authors use Tensor-Train decomposition in order to deal with high-dimensional inputs, obtaining competitive results in comparison to classical approaches for dimensionality reduction. In *Meng & Yang (2021)* and *Yu et al. (2017)* scholars apply tensorization in order to enhance RNN, LSTM models for long-term forecasting of chaotic time series. Using tensors for data representation proved to be effective in natural language problems. Using RNN model with tensor product representation researchers (*Schlag & Schmidhuber, 2018*) managed to beat state-of-the-art models in NLP reasoning tasks. In all considered approaches, despite its effectiveness, applied tensor representations lack interpretability what is the distinguishing feature of the approach proposed in this article.

We should draw a sharp distinction line between the method we use and such popular approach as a Poisson point process (*Chen, Micheas & Holan, 2020*; *Eckardt, González & Mateu, 2021*; *Ghazal & Aly, 2004*; *Komori et al., 2020*; *Marmarelis & Berger, 2005*): we employ not a single Poisson point process, but rather a set of all possible Poisson point processes, squeezed into a single discrete model.

To sum up, despite the variety of methods to represent information about time series one can hardly find the one that re-generates a time series using the corresponding discrete model only, without resorting to observations of the initial time series. Another demand is that it be able to process a time series with a large amount of missing data.

## PROBLEM STATEMENT

For a given set $X$ of time series of the following type: $x = \left( x(t_0), x(t_1), \ldots, x(t_{D_i}) \right)$, where $x(t_i) \in R^K$ ($D_j$ is the length of $j$th time series of $X$; these lengths may be different, but assumed to be large enough) and a given set of discrete models $\Lambda$ that can represent information about a time series, $L : X \to \Lambda$ maps a time series into a discrete model that represents it and $L^* : \Lambda \to X$ generates a time series from a discrete model (the inverse mapping *Campanharo et al., 2011*). The set $\Lambda$ may include either identical models corresponding to different hyperparameters values, or various ones –the only requirement is the ability to implement mappings $L$ and $L^*$ algorithmically. Another considered object is a set of characteristics of a time series $M(x), x \in X$. For a chaotic time series, this set may include the Lyapunov exponents, the generalized entropies, fractal spectra, *etc.* (*Kantz & Schreiber, 2003*; *Malinetskii & Potapov, 2000*; *McCullough et al., 2017*). One can also utilize averaged prediction errors if the algorithm can predict.

The goal is the following: for a given time series $x \in X$ find a model $\lambda^* \in \Lambda$ such that it is a solution for the optimization problem

$$\left\| \frac{M(x) - M\left(L^*(\lambda(x))\right)}{M(x)} \right\| \longrightarrow \min_{\lambda(x)}, \tag{1}$$

provided the number of parameters of $\lambda \in \Lambda$ is less than some threshold $\lambda_{max}$ (defined by the user)

$$|\lambda| \leq \lambda_{max}. \tag{2}$$

The expression $\frac{M(x) - M(L^*(\lambda))}{M(x)}$ symbolizes that the components used to calculate the norm are relative differences between the respective components of the vector $M$ (characteristics of a time series) computed for the initial and re-generated time series. We imply that both mappings does not use the original time series: the mapping $L^* : \Lambda \to X$ depend on the model $\lambda \in \Lambda$ only.

It implies that one attempts to develop a model such that the initial and re-generated time series is as close as possible in terms of a chosen set of time series characteristics $M(x), x \in X$. Comparing the characteristics of the initial time series with those of the re-generated time series, one should use sections of the initial time series that were not used to construct the corresponding model in order to test its ability to generalize, not only to store information. It is possible to seek the minimum of Eq. (1)) calculated either for a separate time series or for a set of time series $X' \subset X$ on average.

## GENERALIZED RELATIONAL TENSOR

Before introducing a formal definition of a generalized relational tensor, we discuss a concept of an observation pattern. A pattern (an irregular time delay embedding scheme) is defined as a pre-set sequence of distances between positions of observations such that these (non-consecutive) observations are to be placed on the consecutive positions in a newly generated sample vector. For example, let us consider a four-point pattern $(2, 3, 4)$. For this pattern, the two first vectors of a training set are $\left(x(t_0), x(t_2), x(t_5), x(t_9)\right)$ and $\left(x(t_1), x(t_3), x(t_6), x(t_{10})\right)$; the last one is $\left(x(t_{m-9}), x(t_{m-7}), x(t_{m-4}), x(t_m)\right)$, where $x(t_m)$ is the last observable value.

The vector, thus concatenated, generalizes a conventional embedding vector (z-vector) (*Kantz & Schreiber, 2003*; *Malinetskii & Potapov, 2000*), which corresponds to the pattern $(1, 1, \ldots, 1)$ ($m$ times). Thus, each pattern is a $S - 1$ - dimension integer vector $(p_1, \ldots, p_{S-1}), p_j \in 1, \ldots, P_{max}, j = 1 \ldots S - 1$; the parameter $P_{max}$ dictates the maximum distance between positions of observations that become consecutive in the vector to be generated. Thereby, the quantity $SP_{max}$ refers to a kind of the memory depth. $\aleph(S, P_{max})$ denotes a set of all possible patterns of the specified length $S$. Figure 1 diagrammatically shows a pattern superimposed on a time series in order to generate a sample vector. This implies that one moves a sliding comb with some teeth broken off (the distance between 'extant' teeth are $p_1, \ldots, p_{S-1}$) along a time series to obtain samples of generalized z-vectors. One should stress that embedding vectors (z-vectors) are usually composed

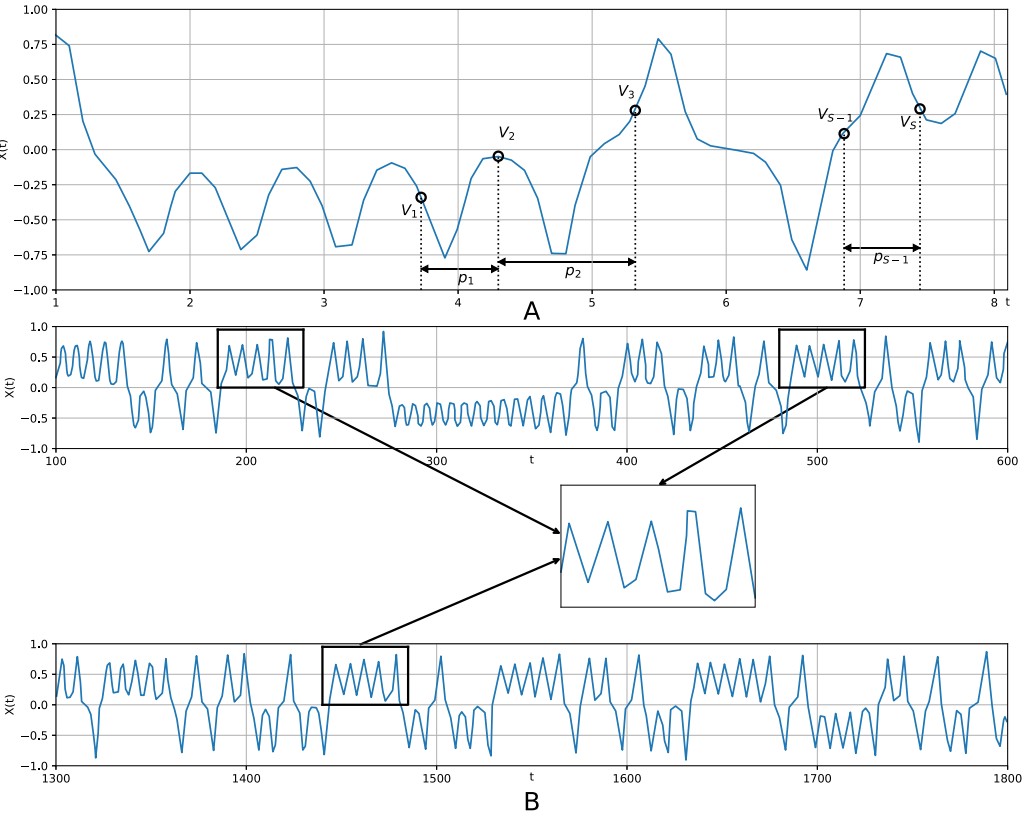

**Figure 1** **A pattern superimposed on the Lorenz time series in order to generate a sample vector.** (A) shows a motif $(v_1, v_2, \ldots, v_s)$; generated according to a pattern $(p_1, p_2, \ldots, p_{s-1})$; (B) illustrates how a motif is determined: similar sections from various parts of a time series are averaged to create a motif.

of consecutive observations. Generalized z-vectors are composed of non-consecutive observations according to a certain pattern (irregular embedding scheme).

The generalized z-vectors proved to be efficient prediction tools in the framework of the predictive clustering approach (*Aghabozorgi, Shirkhorshidi & Wah, 2015*; *Blockeel, Raedt & Ramong, 1998*; *Martinez Alvarez et al., 2011*). Predictive clustering implies that in order to predict a given position, one should seek in time series for sequences similar to that immediately preceding the position in question. The final observations of such sequences are used as predictions. To implement this idea, one clusters all possible short sections of the time series and utilizes centers of the clusters (motifs) as prediction tools. *Gromov & Borisenko (2015)* show that if one employs motifs corresponding to various patterns (generalized z-vectors), one may essentially improve prediction quality. Hence, the discrete model that in point of fact compresses information about such motifs has aroused considerable interest.

The algorithm considered in the present article, in contrast to known methods used to reveal motifs in time series, does not store the motifs separately, but compresses them into the tensor in order to reduce the number of parameters stored.

To define the generalized relational tensor, we assume that the time series observations belong to the interval $[-1, 1]$. The interval is divided into $N$ equal subintervals; all observations of the time series are $K$-dimensional.

If elements of an observation $x(t_i)$ belong to intervals $i_1, i_2, \ldots, i_K, i_k \in 1 \ldots N, k = 1 \ldots K$, then it corresponds to an index vector $I = (i_1, i_2, \ldots, i_K)$. A set of (non-consecutive) observations then yields a concatenation of $S$ $K$-dimensional vectors $I_j = (i_{j1}, i_{j2}, \ldots, i_{jK})$.

**Definition 1**

*The tensor $T \in \Lambda$ is said to correspond to the set of all patterns of the length $S\aleph(S, P_{max})$ and to $K$-dimensional observations, if its multi-index is a concatenation of $SK$-dimensional vectors $I_j = (i_{j1}, i_{j2}, \ldots, i_{jK})$ and a $S-1$-dimensional vector $P \in \aleph(S, P_{max}) : I = \bigcup_{j=1}^{S} I_j \bigcup P$. Indices $i_{jk}, j = 1 \ldots S, k = 1 \ldots K$ take the values $i_{jk} \in 1, \ldots, N$.*

**Definition 2**

*The generalized relational tensor $T$ is the tensor $T$ corresponding to $\aleph(S, P_{max})$ such that if one assigns to its multi-index values $I^* = (i_{jk} = i_{jk}^*, p_j = p_j^*)$, then one considers $S$ observations with positions separated by distances $p_1^*, p_2^*, \ldots, p_{S-1}^*$ and respective values belonging to the subintervals $i_{1k}^*, i_{2k}^*, \ldots, i_{Sk}^*, k = 1 \ldots N$. The value of the respective tensor element is an estimated probability to encounter (in the time series in question) observations that belong to intervals $I_j^* = (i_{j1}^*, \ldots, i_{jK}^*)$ at positions separated by distances $p_1^*, p_2^*, \ldots, p_{S-1}^*$, respectively.*

In the predictive clustering one does not compress motifs into a single generalized relational tensor, but rather considers all clusters for all patterns (as in *Gromov & Borisenko, 2015*). Respectively, each cluster serves as a simple statistical model, associated with several time series sections (to put it differently, with a certain region of its strange attractor). A set of values of a particular component of cluster elements corresponds to a sample probability density function for a certain random variable, and the cluster itself encodes statistical relations among such variables. Naturally, the motifs (clusters' centers) with lengths $S > 2$ make sense only if the time series in question is essentially non-Markovian. In the present article, we consider relational tensors, which compress the information stored in the motifs into a single object. The tensor naturally comprises significantly fewer parameters at the cost of partially lost information. On the other hand, if one sets $S = 2$, then the relation tensor becomes a conventional transition matrix of a Markov chain.

## METHOD TO GENERATE GENERALIZED RELATIONAL TENSOR

To generate a generalized relational tensor for a time series, we employ modified ant colony optimization (*Dorigo & Gambardella, 1997*). The algorithm imitates the way ant colonies forage for food items in the wild. The central idea of the algorithm is to move in a search space and increase weights of all edges in the path. If the path appears good, then the ant deposits a certain amount of pheromone along the path directly proportional to the goodness of the path. The probability for an ant to select an edge is directly proportional to the amount of pheromone already deposited. This algorithm combines probabilistic search, typical for evolutionary algorithms, with rather high speed of search, typical for

classical optimization techniques. The brute force algorithm to generate the tensor is computationally prohibitive.

The parameters of the algorithm in question, besides $N, S, P_{max}$, and $K$ described above, are the following:

- The initial amount of pheromone deposited on all elements of the tensor, $p_0$.
- The number of transitions that an ant makes before it deposits pheromone for the next time (during a single step of the algorithm), $Q_1, Q_1 \leq S-1$; usually, $Q_1 = S-1$.
- The amount of pheromone that an ant deposits along the completed path, $\Delta p$.
- The total number of transitions that an ant can accomplish as it moves along a time series, $Q_2, Q_2 \gg Q_1$. Usually, $Q_2 \sim D$ and, therefore, an ant does not stop moving unless it reaches the end of the time series.
- The amount of pheromone evaporating from all tensor elements (all tensor elements are decreased by the same value) after each step, $-\Delta p_a (\Delta p_a \ll \Delta p)$.

The procedure *Quantify* for a set of (normalized) real-valued numbers $x_{ik}, k = 1 \ldots K$, returns a set of intervals to which they belong. The procedure *ConstTensor(a)* fills all elements of the tensor with a given value a; the operator $|T|_+$ replaces all negative elements of the tensor $T$ with zeroes.

For the algorithm to generate the relation tensor, a time series serves as input data, while a filled generalized relational tensor is its output data.

**Algorithm 1 (to generate a generalized relational tensor)**

All elements of the tensor T are filled with the value $p_0$ before the first step:

$T \leftarrow ConstTensor(p_0)$

A step of the algorithm.

1. A random position in the time series is selected ($t_0 \leftarrow Random, t \leftarrow t_0$).
2. An ant makes $Q_1$ transitions. For arbitrary $q, q = 1 \ldots Q_1$, it makes the following:
   2.1 $I_q^* \leftarrow Quantify(x(t))$
   2.2 The next position is chosen probabilistically where the probability to transit to (leap for) $p_q^*$ positions ahead is determined as

$$P(p_q = p_q^*) =$$
$$= \frac{\sum_{(I_{q+1}, \ldots, I_S, p_{q+1}, \ldots, p_{S-1})} T\left(I_1^*, \ldots, I_q^*, I_{q+1}, \ldots, I_S, p_1^*, \ldots, p_{q-1}^*, p_q^*, p_{q+1}, \ldots, p_{S-1}\right)}{\sum_{(I_{q+1}, \ldots, I_S, p_{q+1}, \ldots, p_{S-1})} T\left(I_1^*, \ldots, I_q^*, I_{q+1}, \ldots, I_S, p_1^*, \ldots, p_{q-1}^*, p_q, \ldots, p_{S-1}\right)}; t \leftarrow t + p_q. \quad (3)$$

3. After $Q_1$ transitions, the algorithm performs the following operations:
   3.1 A pheromone is deposited along an ant's path

$$T\left(I_1^*, \ldots, I_S^*, p_1^*, \ldots, p_{S-1}^*\right) \leftarrow T\left(I_1^*, \ldots, I_S^*, p_1^*, \ldots, p_{S-1}^*\right) + \Delta p. \quad (4)$$

   3.2 A pheromone evaporates from all elements of the tensor

$$T \leftarrow |T - ConstTensor(\Delta p_a)|_+. \quad (5)$$

After that, a new ant is placed randomly in this time series.

Possible termination criteria can be defined by:

1. The maximum total number of ants' steps along time series considered.
2. Small overall rate of changes of the tensor elements during several consecutive steps.

To generate a time series with the employment of a given generalized relational tensor, we employed the Algorithm 2. (It is suggested that $S-1$ random observations of the initial time series are taken as initial conditions.) The parameter of the algorithm in question is $Q_3$ –the total number of transitions an ant can make before it is replaced by another ant.

---

**Algorithm 1** Generation of a generalized relational tensor

1:   **function** GGRT$(x(t), I, Q_1)$
2:     $T \leftarrow ConstTensor(p_0)$
3:     **while not** termination condition **do**
4:       $t_0 \leftarrow Random,\ t \leftarrow t_0$
5:       **for** $q = 1, \ldots, Q_1$ **do:**
6:         $I_q^* \leftarrow Quantify(x(t))$
7:         $P(p_q = p_q^*) = \dfrac{\sum_{I_{q+1},\ldots,I_S,p_{q+1},\ldots,p_{S-1}} T\left(I_1^*,\ldots,I_q^*,I_{q+1},\ldots,I_S,p_1^*,\ldots,p_q^*,p_{q+1},\ldots,p_{S-1}\right)}{\sum_{I_{q+1},\ldots,I_S,p_{q+1},\ldots,p_{S-1}} T\left(I_1^*,\ldots,I_q^*,I_{q+1},\ldots,I_S,p_1^*,\ldots,p_q,p_{q+1},\ldots,p_{S-1}\right)}$
8:         $t \leftarrow t + \Delta p_0$
9:     $T\left(I_1^*,\ldots,I_S^*,p_1^*,\ldots,p_{S-1}^*\right) \leftarrow T(I_1^*,\ldots,I_S^*,p_1^*,\ldots,p_{S-1}^*) + \Delta p$
10:    $T \leftarrow |T - ConstTensor(\Delta p_a)|_+$
     **return** $T$

---

**Algorithm 2 (to generate a time series)**

1. An ant is placed on the tensor element defined by the initial conditions (for the first ant) or by already filled positions (for all other ants). In the latter case, from a set of already filled observations, the algorithm selects the one that immediately precedes the first unfilled one. It becomes the next start position.
2. Both position to which an ant transits and the interval to which the value of respective observation belongs are determined probabilistically, using the roulette wheel, with probabilities determined as

$$P\left(I_S = I_S^*, p_{S-1} = p_{S-1}^*\right) = \frac{T\left(I_1^*,\ldots,I_S^*,p_1^*,\ldots,p_{S-2}^*,p_{S-1}^*\right)}{\sum_{(I_S,p_{S-1})} T\left(I_1^*,\ldots,I_{S-1}^*,I_S,p_1^*,\ldots,p_{S-2}^*,p_{S-1}\right)}. \qquad (6)$$

3. If the position selected is already filled, then go to 1, else continue $t \leftarrow t + p_S$.
4. The ant moves till the distance between the first position it took and the last generated is lesser than $Q_3$.
5. If some time series positions are unfilled, then place a new ant to the position of an already filled observation that immediately precedes the first unfilled one and go to 1.

The algorithm outlined above gives the number of an interval only. One can calculate the concrete value as the center of the interval, a realization of the random variable uniformly distributed over this interval, or averaged observations that belong to this interval. The last option appears to be the best, but it requires to calculate the averaged values that these averaged values be calculated when the tensor is generated.

---

**Algorithm 2** Generation of time series from tensor

---

1: $Y = \{y_t\}_{t=1}^{D}, y_t = [\emptyset]^K$        $\triangleright$ $y_t$ is an empty $K$ dimensional array

2: $\tilde{I}_j \leftarrow Quantify(x_{t_j}), \forall j=1,\ldots,S-1; x_{t_1},\ldots,x_{t_{S-1}} \in X$        $\triangleright$ initial condition

3: $Y_j \leftarrow \tilde{I}_j, \forall j=1,\ldots,S-1$

4: **function** GTS$(T, Y, Q_3)$

5:      **while** $[\emptyset]^K \subset Y$ **do**

6:          $antway = 0$

7:          **while** $antway < Q_3$ **do**

8:             take $t_1, t_2, \ldots, t_{S-1} \subset \arg\left(\{\{y_t\}_{t=1}^{D} | y_j \neq [\emptyset]^K, \forall j \in 1,\ldots,D\}\right)$

9:               $\triangleright$ select the one that immediately precedes the first unfilled one according to probability

10:             $I_1, \ldots, I_{S-1} \leftarrow y_{t_1}, \ldots, y_{t_{S-1}}$

11:             $p_1, \ldots, p_{S-2} \leftarrow t_2 - t_1, \ldots, t_{S-1} - t_{S-2}$

12: 
$$P\left(I_S = I_S^*, p_{S-1} = p_{S-1}^*\right) = \frac{T\left(I_1^*, \ldots, I_S^*, p_1^*, \ldots, p_{S-2}^*, p_{S-1}^*\right)}{\sum_{(I_S, p_{S-1})} T\left(I_1^*, \ldots, I_{S-1}^*, I_S, p_1^*, \ldots, p_{S-2}^*, p_{S-1}\right)}$$

13:             $(I_S^*, p_{S-1}^*) \sim P(I_S^*, p_{S-1}^*)$

14:             **if** $y_{t_{S-1}+p_{S-1}^*} = [\emptyset]^K$ **then**

15:                $y_{t_{S-1}+p_{S-1}^*} \leftarrow I_S^*$

16:                $antway \leftarrow antway + p_{S-1}^*$

     **return** $Y$

---

We should stress that while the number of parameters is comparatively large, for the most of them there exists a kind of 'default' values in the ACO theory, and we did not attempt to perturb them. What actually works is a pair $S, P_{max}$ that determines the lengths of sliding combs, an initial amount of pheromone $p_0$, its rate of evaporation $-\Delta p_a$, and the number of ants. As for the first parameter $S$, it should be equal to 3 or 4, since smaller values are not sufficient to allow definite conclusions for chaotic time series, and larger values lead to combinatorial explosion. $P_{max}$ (a maximum number of steps between neighboring comb teeth) is reasonable to choose near 10 in order to grasp typical motifs of time series; $(S-1)P_{max}$ is a kind of memory depth. $p_0$ and $-\Delta p_a$ are not independent parameters; their ratio should be large enough to allow ants to make a large number of movements. An alternative strategy implies that one does not apply evaporation operation ($\Delta p_a = 0$), but subtract $p_0$ when all ants' movements are finished. As for the number of ants, the more is the better.

The filled generalized relational tensor can be used for time series forecasting. As the predicted value for the position $t$ one chooses the most probable $I_S$: $T(I_1^*, \ldots, I_{S-1}^*, I_S, p_1, \ldots, p_{S-1})$ (with $I_{S-q}^* = Quantify(x(t - \sum_{j=1}^{q} p_{S-j})), q = 1 \ldots S-1$) out of all possible patterns of $\aleph(S, P_{max})$. The tensor remembers the most common sequences occurring in the time series (the most frequently visited areas by ants) and stores them in its elements. The more often a sequence occurs in a series, the more likely this sequence will be in the tensor.

The algorithms prove to be efficient, provided that transient processes are completed, and the time series is associated with trajectories belonging to the neighborhood of a (strange) attractor, or the time series is $\varepsilon$-stationary (*Orlov & Osminin, 2011*).

## RESULTS

The algorithms described above were applied to generate generalized relational tensors for a noisy periodic time series, the standard chaotic time series (generated by the Lorenz system and by Mackey-Glass equations), and the real-world chaotic data (hourly load values in Germany, from 23:00 12/31/2014 to 14:00 20/02/2016, https://www.entsoe.eu/data/power-stats/).

For each considered series, we calculate the characteristics of the original series and the series restored from the tensor. We consider the problem successfully solved if these characteristics match *i.e.,* the value of the functional (1) is close to zero.

Figures 2A and 2B show the initial non-noisy periodic time series (with the step length equal to $h = 0.1$) and the one that was generated by its generalized relational tensor. Training and testing sets comprise 100,000 and 10,000 observations, respectively.

Normally distributed noise, with mean 0 and variance $\xi = 0.5$, is added to observations of the initial periodic time series. Figure 3 graphically exhibits average squared error *vs.* pattern length $S$ for $N = 20, K = 1, p_0 = 0.1, \Delta p = 0.1, \Delta p_a = 0.0001$.

Obviously, there is an almost complete coincidence of the original and restored series, the better, the longer are the routes the ants pass (and store information about them into the tensor) (Fig. 3). Furthermore, the algorithm was checked on the Lorenz time series. The Lorenz time series is defined by the following system of differential equations:

$$\begin{cases} \dot{x}_1 & = \sigma (x_2 - x_1) \\ \dot{x}_2 & = x_1 \sigma (r - x_3) - x_2 \\ \dot{x}_3 & = x_1 x_2 - b x_3 \end{cases} \tag{7}$$

where its parameters take conventional chaotic values $\sigma = 10, r = 28, b = 8/3$.

To obtain the time series, the Lorenz system is integrated using the fourth-order Runge–Kutta method with the integration step is equal to $h = 0.1$. Training and testing sets comprise 100,000 and 10,000 observations, respectively.

For chaotic time series the pointwise comparison of the initial and re-generated time series is irrelevant. To assess the tensor, we compare the following characteristics of these time series: partial auto-correlation functions, the largest Lyapunov exponents, spectra of generalized entropies, and entropy-complexity pair. To estimate the largest Lyapunov exponent, the TISEAN package (*Kantz & Schreiber, 2003*; *Malinetskii & Potapov, 2000*) is used. The spectrum of the generalized entropies is computed with the employment of the method based on the box-counting (*Kantz & Schreiber, 2003*; *Malinetskii & Potapov, 2000*). To compare, *Malinetskii & Potapov (2000)* (see also *Kantz & Schreiber, 2003*) estimate the largest Lyapunov exponent for this time series as $\lambda_1 = 0.91$. The monograph also discusses its full Lyapunov spectrum and the spectrum of the generalized entropies. To calculate entropy-complexity pair, we use the method discussed in *Martin, Plastino & Rosso (2006)* and *Rosso et al. (2007b)*.

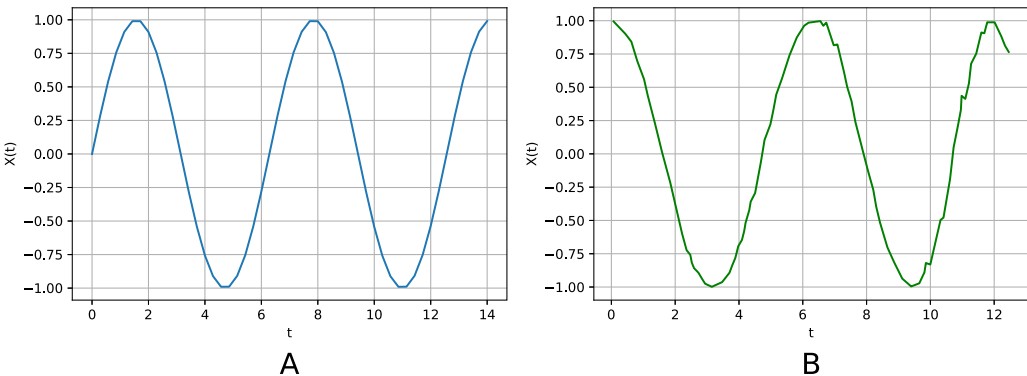

**Figure 2** **The initial (non-noisy) periodic time series (A) and the one generated by the respective relational tensor (B) (noise variance $\xi = 0.5$).** The centers of the intervals are used as the values of the generated time series. The average squared error is $\varepsilon = 0.02$.

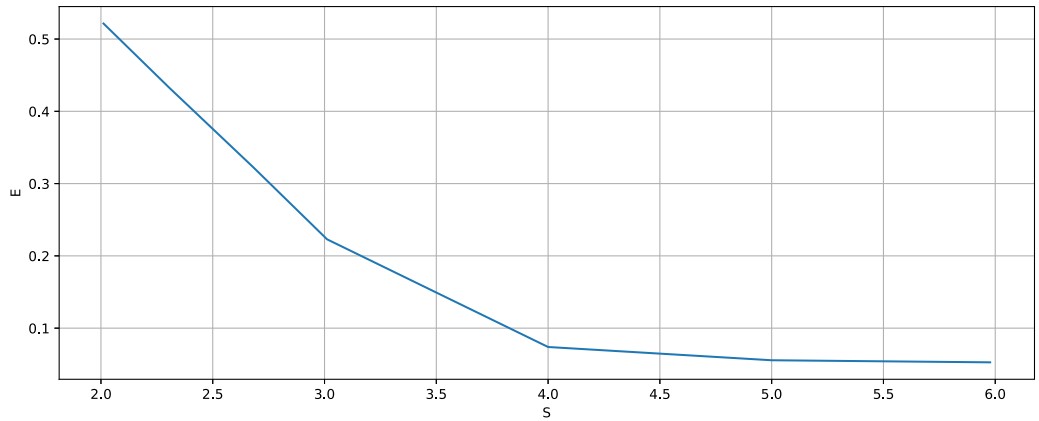

**Figure 3** **Noisy periodic time series.** Average squared error *vs.* pattern length $S$. $N = 20$.

[4]Obviously, there can no longer be a pointwise coincidence, as it was for the sine.

Figure 4 shows the results corresponding to the initial Lorenz time series (the left column) and the one generated by the respective generalized relational tensor (the right column). Figures 4A, 4B show typical sections of the time series.[4] Figures 4C and 4D display partial auto-correlation functions; Figs. 4E and 4F present the spectra of the generalized entropies. The figure corresponds to the optimal values $K = 3$, $N = 200$, $P_{max} = 2$, $S = 4$, $p_0 = 0.1$, $\Delta p = 0.1$, $\Delta p_a = 0.00001$.

The calculated largest Lyapunov exponent for the initial time series is $\lambda_1 = 0.9$, while for the re-generated time series it is $\lambda_1 = 0.91$, thus the relative error does not exceed 1.2%. The relative error for the first ten points of the auto-correlation function amounts to 21.8%, for the spectra of the generalized entropies the error equals to 3%. In particular, the estimated KS-entropies of the initial and re-generated time series are equal to $K_2 = 0.92$ and $K_4 = 0.94$, respectively. The entropy-complexity pair for the original time series is $(0.53; 0.43)$; for the re-generated one $(0.51; 0.43)$. The relative errors are, respectively, 3% and 0.08%.

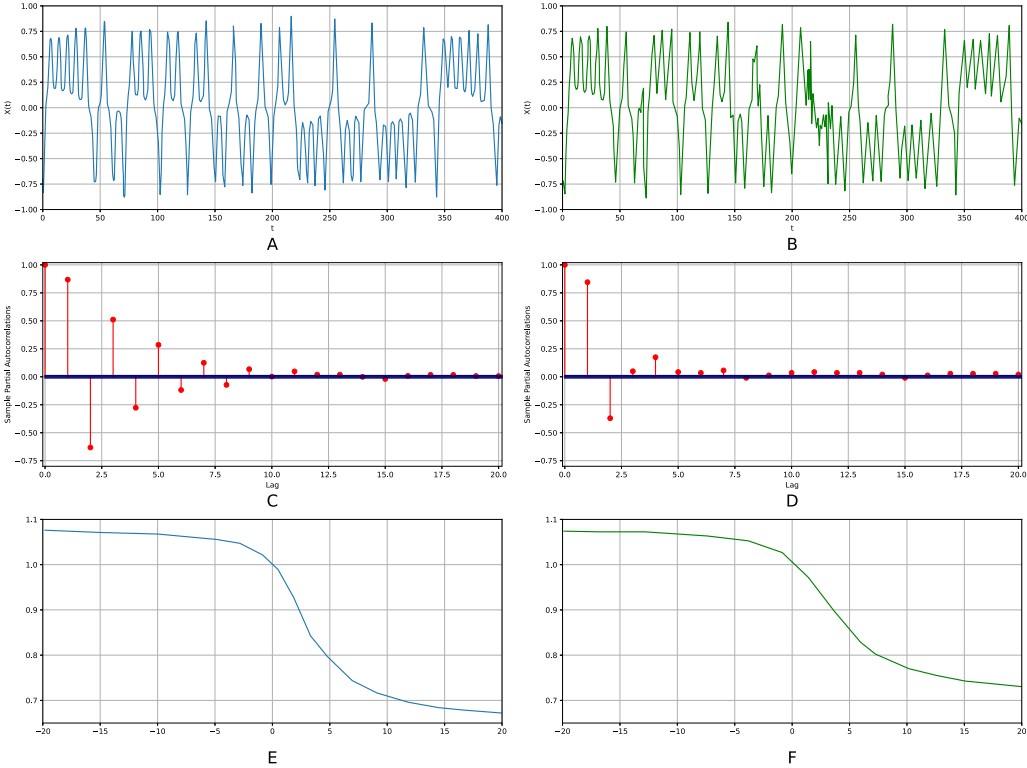

**Figure 4** **The initial Lorenz time series (the left column) and the one generated by the respective relational tensor (the right column).** (A and B) Show typical sections of the time series; (C and D) display partial auto-correlation functions; (E and F) present the spectra of the generalized entropies. $K = 3$, $N = 200$, $P_{max} = 2$, $S = 4$, $p_0 = 0.1$, $\Delta p = 0.1$, $\Delta p_a = 0.00001$.

Figure 5 shows a typical section of the Lorenz time series and the respective predicted values for 10 steps ahead prediction. The average relative prediction error amounts to 35%. The results are considered to be satisfactory for multi-step ahead prediction due to the exponential growth of an error intrinsic to chaotic time series.

To examine the algorithm as a means to process irregularly sampled data, 10% of randomly chosen observations is dropped from the sampled Lorenz time series. We use several dropping patterns corresponding to various probability distributions (see *Sakellariou et al., 2016* for details). Table 1 comprises MPEs for entropy, complexity and the largest Lyapunov exponent as well. It makes it possible to conclude that the proposed tensor can regenerate time series with missing observations.

The scalability of the algorithm seems to be of fundamental importance. In order to check it, we performed a large-scale simulation for a time series with length ranging from $10^5$ to $10^{10}$. Figure 6 displays the respective results. A logarithm of a sample size is plotted on the abscissa axis for all three subfigures. On the ordinate axis, Fig. 6A displays the number of non-zero elements of a tensor; Fig. 6B a relative error for the largest Lyapunov exponent; Fig. 6C a relative error for the position on the entropy-complexity plane. Figure 6A shows that the number of non-zero tensor elements grows exponentially. The fact conforms to

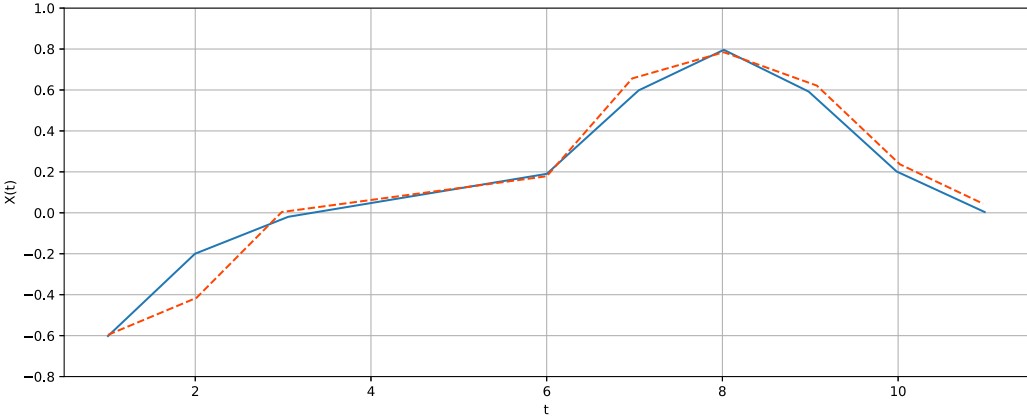

**Figure 5** Typical section of the Lorenz time series (blue solid line) and the respective predicted values (orange dashed line) for 10 steps ahead prediction.

**Table 1** Uniform and non-uniform sampling. Discrepancies between characteristics of the original and re-generated time series.

| Sampling | Entropy, MPE % | Complexity, MPE % | HLE (TISEAN), MPE % |
|---|---|---|---|
| Uniform | 3 | 0.08 | 1.92 |
| Poisson | 15.5 | 1.3 | 5.48 |
| Pareto | 11.5 | 3.9 | 14.52 |
| Gamma | 15.4 | 1.3 | 14.53 |

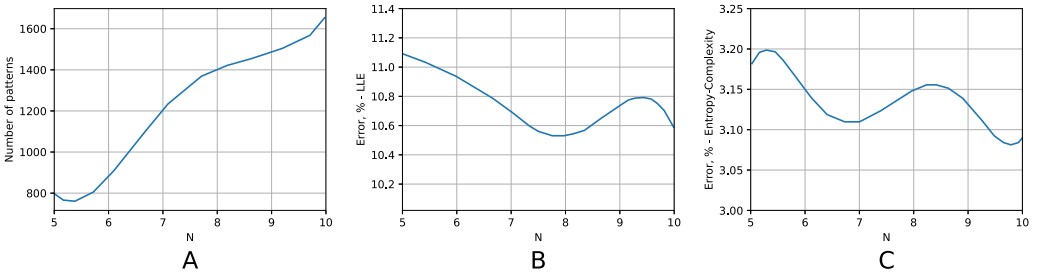

**Figure 6** Scalability of the algorithm. The Lorenz time series. On the abscissa axis, for all three subfigures, a logarithm of a sample size is plotted (the sample size ranges from $10^5$ to $10^{10}$). On the ordinate axis, (A) displays the number of non-zero elements of a tensor; (B) a relative error for the largest Lyapunov exponent; (C) a relative error for the position on the entropy-complexity plane.

the chaotic nature of the time series. The greater is sample size, the more probability for the respective trajectory section to visit rarely visited regions of a strange attractor (regions with small invariant measure). Consequently, the more typical sequences (motifs) are added to the tensor. Meanwhile, the relative error is nearly constant for increasing sample

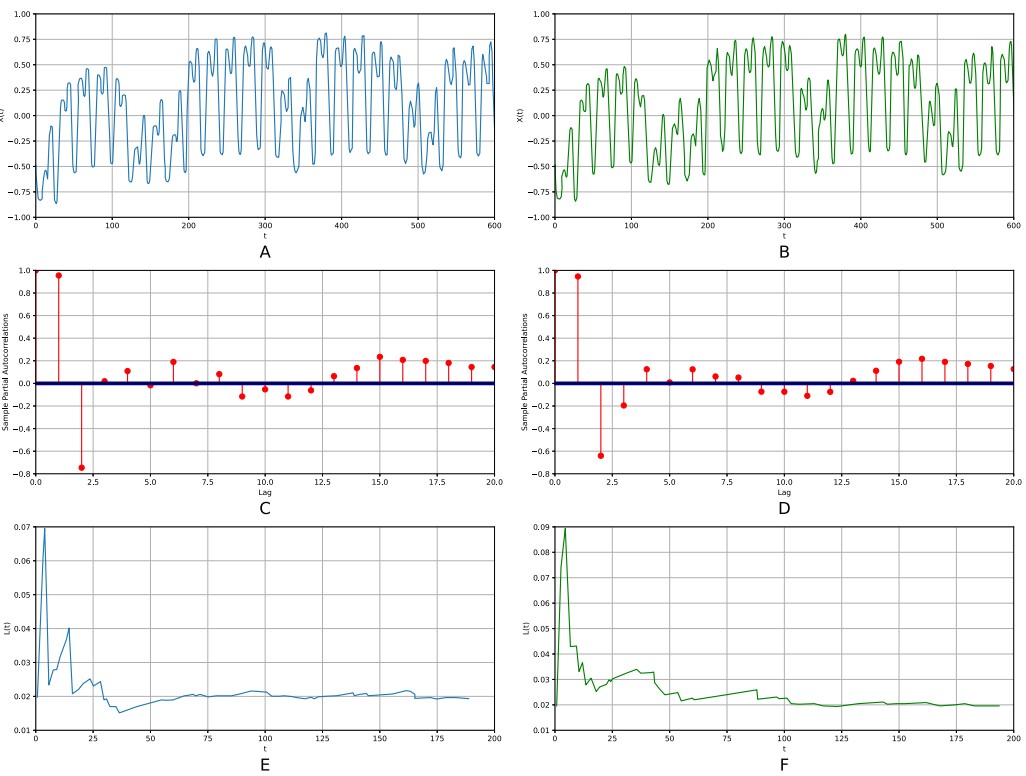

**Figure 7** **The initial German hourly load time series and the one generated by the respective relational tensor.** (A and B) Show typical sections of the time series; (C and D) display partial auto-correlation functions, (E and F) depict convergence of approximations to the largest Lyapunov exponents (the Rosenstein method). $K = 3$, $N = 200$, $P_{max} = 2$, $S = 4$, $p_0 = 0.1$, $\Delta p = 0.1$, $\Delta p_a = 0.00001$.

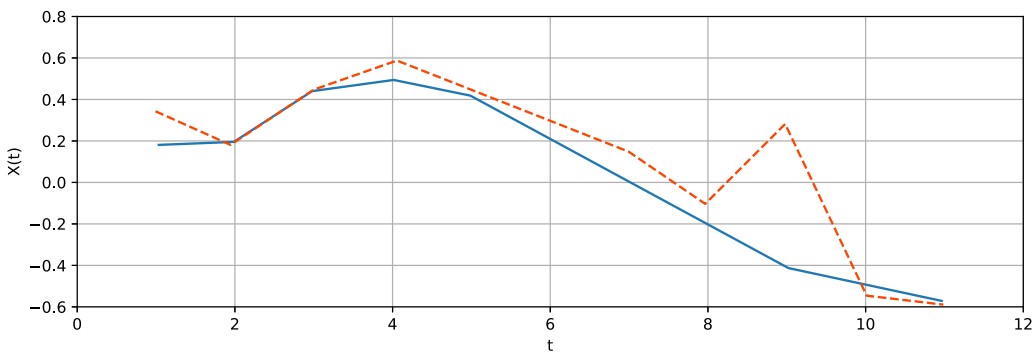

**Figure 8** **Typical section of energy consumption time series (blue solid line) and the respective predicted values (red dashed line) for 10 steps ahead prediction.**

size, starting from a certain threshold. Quite clearly, that there is no need to enlarge the sample size over this threshold and the algorithm is scalable in this sense.

Another issue of importance is how hyper-parameters affect algorithm efficiency. To address the problem, we also performed a wide-ranging simulation. Resulting plots are

presented in Appendix A (see Figs. A1–Fig. A6). It was ascertained that the number of tensor dimensions $S$ weakly influences the algorithm performance. The optimum value amounts to 3. A value of $Q$ also scarcely affects the algorithm performance. Vice versa, the number of intervals of the range of values $N$ strongly affects the performance. For example, for the Lorenz time series, as the number of intervals ranges from 20 to 100, the relative error for the largest Lyapunov exponent ranges from 14% to 25%, and its counterpart for the position on the entropy-complexity ranges from 3% to 10%. The best value for all time series under study appears to be 60 intervals. The algorithm performance also depends strongly upon pheromone evaporation $\Delta p_a$, with the best value equal to $10^{-6}\Delta p$. Pheromone change $\Delta p$ and $P_{max}$ affect it fairly weakly. The algorithm performance was also tested on the Mackey-Glass chaotic time series. Similar results were obtained.

Finally, the algorithm was applied to analyse chaotic real-world data: energy consumption in Germany from 23:00 12/31/2014 to 14:00 20/02/2016; (http://www.entsoe.eu/data/power-stats). The time series is chaotic as its largest Lyapunov exponent is positive ($\lambda_1 = 0.12$). The value was estimated using the TISEAN package. Figure 7 shows the results corresponding to the initial time series (the left column) and the one generated by the respective generalized relational tensor (the right column). Figures 7A and 7B show the typical sections of the time series; Figs. 7C and 7D display partial auto-correlation functions. The figure corresponds to the same values $K = 1$, $N = 200$, $P_{max} = 2$, $S = 4$, $p_0 = 0.1$, $\Delta p = 0.1$, $\Delta p_a = 0.00001$. The largest Lyapunov exponents for the initial and re-generated time series are $\lambda_1 = 0.12$ and $\lambda_1 = 0.18$ respectively. An entropy-complexity pairs are $(0.5; 0.38)$ and $(0.53; 0.37)$ that yields relative errors 4.8% and 2.1%, respectively.

The average relative error for the first ten points of the auto-correlation function is 20.5%. In order to test the algorithm's ability to generalize, we compared values of these results with those calculated for a section of the time series that has not been used to fill the tensor—the respective errors appear to be 7.2% and 19.7%.

Figure 8 shows the typical section of this time series and the respective predicted values for 10 steps ahead prediction. The average relative prediction error amounts to 45%.

In actual practice, the most frequent motifs makes it possible to examine typical patterns in energy consumption. The set of patterns allows designing an appropriate strategy for energy generation. Such strategy is of vital importance for the concept of a smart city. The concept suggests that one should balance energy generation and consumption. As it is hardly possible to change energy generation quickly, one should examine such patterns, in order to offset the energy generation and consumption. Most important are the patterns which precede blackouts.

We would like to discuss the limitations of the proposed method. The method implies that one seeks for motifs of the time series in question and aggregates them in a single discrete model. Both advantages and disadvantages of the method rely on this fact. The algorithm proves efficient provided data are not so noisy. On the controversy, if the noise amplitude is large, the number of non-zero tensor elements grows exponentially, and the algorithm performance deteriorates.

On the completion, we would like to elaborate on potential applications and future directions. The discrete model that represents chaotic time series with a guarantee that its chaotic characteristics are preserved has lots of potential applications. First of all, it seems reasonable to examine these models corresponding to time series readily interpretable in the context of their subject areas (biomedical, energy, weather time series, and so forth). In such a case such model may help reveal hidden relations and relate them to the laws that govern the subject area. In particular, it seems possible to identify time series belonging to different classes with the employment of the corresponding tensors. The second possible direction is to adjust the tensor understudy and modify appropriate algorithms in order to tackle time series with a pronounced noise term (financial, for example).

Finally, the third direction is to predict many steps ahead. It is worth noting that there are several fairly efficient methods to predict to one step ahead. Unfortunately, when it comes to many steps, the prediction methods are hardly available due to the Lyapunov instability of trajectories of chaotic dynamical systems. Meanwhile, some articles (associated mainly with predictive clustering) that discuss many-steps prediction methods (*Gromov & Borisenko, 2015*) make it possible to expect new results in this complex and important problem. The new algorithm offers to compose motifs from non-consecutive observations according to certain patterns. The main drawback of this approach is an exponential growth of the number of motifs and, consequently, of computation time. From this angle, the relation tensor squeezes all these motifs into a relatively compact structure. Of course, it partially loses pieces of information and deteriorates prediction quality. It would be interesting to estimate this deterioration for real-world time series.

## CONCLUSIONS

The central idea of this work is the assertion that the only correct way to assess the quality of the representation of a time series (no matter how this representation is formed) is the proximity of the characteristics of the original and restored series. (The Lyapunov spectrum, dimensions of attractors, entropy-complexity, partial autocorrelation function, *etc.*, can serve as such characteristics.) The second goal that we pursued in this article is to show that the solution of the problem in such formulation is possible: we have presented a new type of discrete model and algorithms for filling/restoring it, which meets the specified requirements.

For simple deterministic series, there is a complete coincidence of the original and restored series. For chaotic time series, a difference between characteristics of the original time series (the largest Lyapunov exponent, the auto-correlation function, pair of entropy, and complexity) and those of the time series re-generated from a model is used to assess the effectiveness of the algorithm in question. The approach has shown fairly good results for periodic and benchmark chaotic time series and satisfactory results for real-world chaotic data.

The model allows readily processing irregularly sampled time series.

We study the dependence of the algorithm performance on its parameters; the algorithm appears to be rather robust.

# APPENDIX

In the following figures, the blue curves correspond to the Lorenz time series; the orange curves, to the Mackey-Glass time series; the green curve, to energy consumption time series (see above).

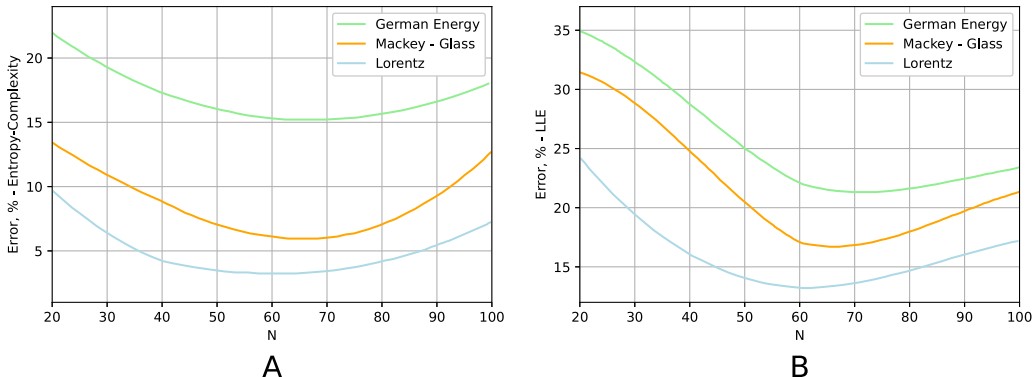

**Figure A1** A relative error for the position on the entropy-complexity plane (A) and for the largest Lyapunov exponent (B) for different values of the parameter $N$. $P_{max} = 2, S = 3, \Delta p = p_0, \Delta p_a = 10^{-7} \Delta p, Q = S - 1$.

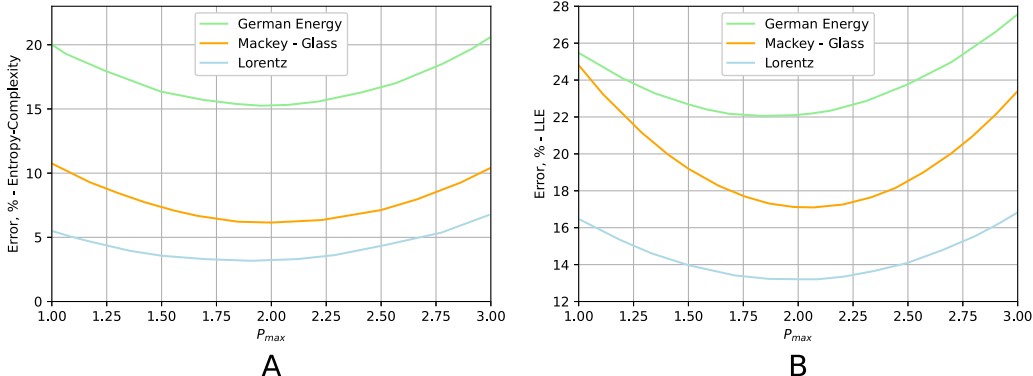

**Figure A2** A relative error for the position on the entropy-complexity plane (A) and for the largest Lyapunov exponent (B) for different values of the parameter $P_{max}$. $N = 60, S = 3, \Delta p = p_0, \Delta p_a = 10^{-7} \Delta p, Q = S - 1$.

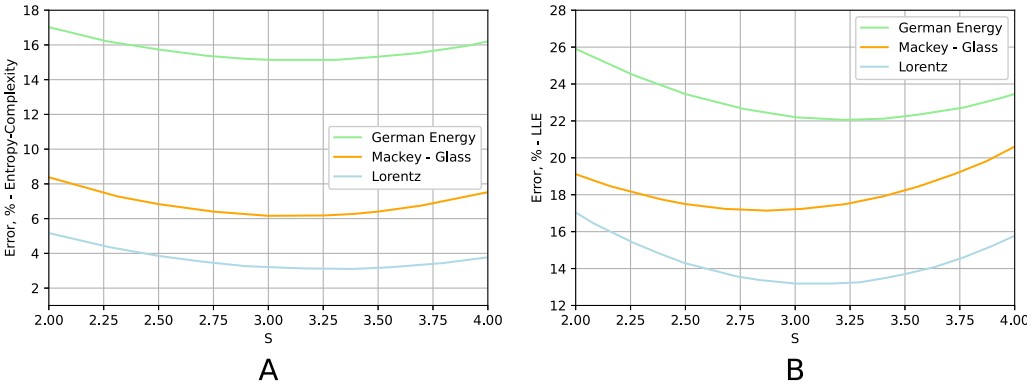

**Figure A3** **A relative error for the position on the entropy-complexity plane (A) and for the largest Lyapunov exponent (B) for different values of the parameter S.** $N = 60, P_{max} = 2, \Delta p = p_0, \Delta p_a = 10^{-7}\Delta p, Q = S - 1.$

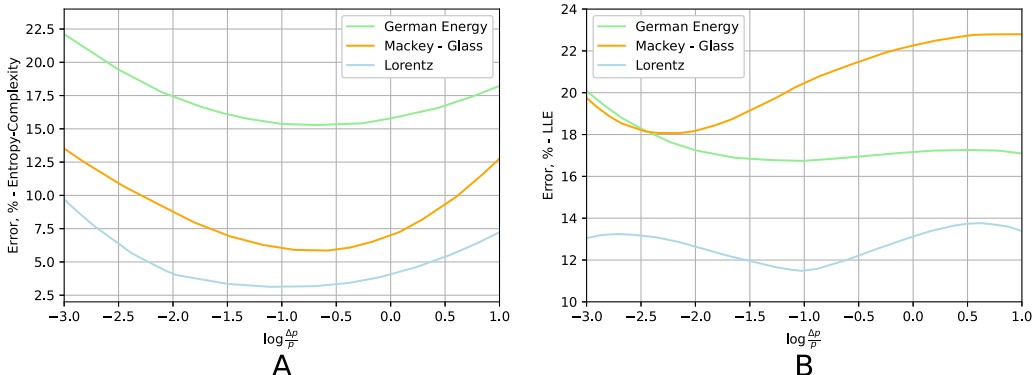

**Figure A4** **A relative error for the position on the entropy-complexity plane (A) and for the largest Lyapunov exponent (B) for different values of the parameter $\Delta p$.** $N = 60, P_{max} = 2, S = 3, \Delta p_a = 10^{-7}\Delta p, Q = S - 1.$

### Funding

The publication was supported by the grant for research centers in the field of AI provided by the Analytical Center for the Government of the Russian Federation (ACRF) in accordance with the agreement on the provision of subsidies (identifier of the agreement 000000D730321P5Q0002) and the agreement with HSE University No. 70-2021-00139.

### Grant Disclosures

The following grant information was disclosed by the authors:
Analytical Center for the Government of the Russian Federation (ACRF): 000000D730321P5Q0002.

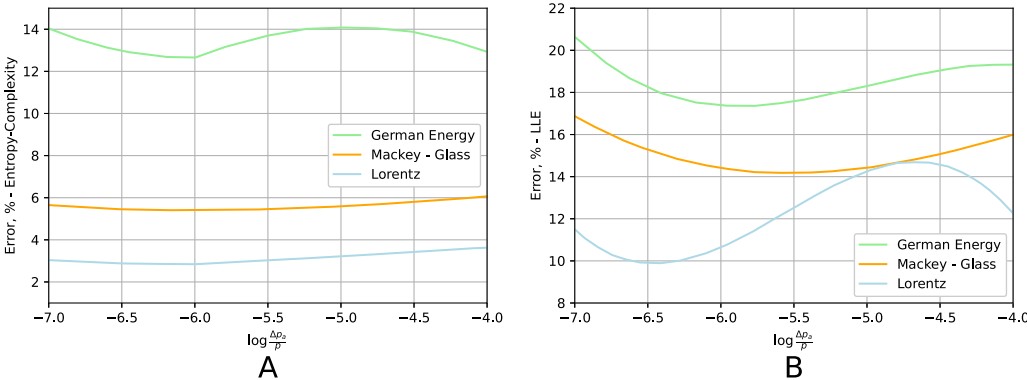

**Figure A5** A relative error for the position on the entropy-complexity plane (A) and for the largest Lyapunov exponent (B) for different values of the parameter $\Delta p_a$. $N = 60$, $P_{max} = 2$, $S = 3$, $\Delta p_a = 10^{-7}\Delta p$, $Q = S - 1$.

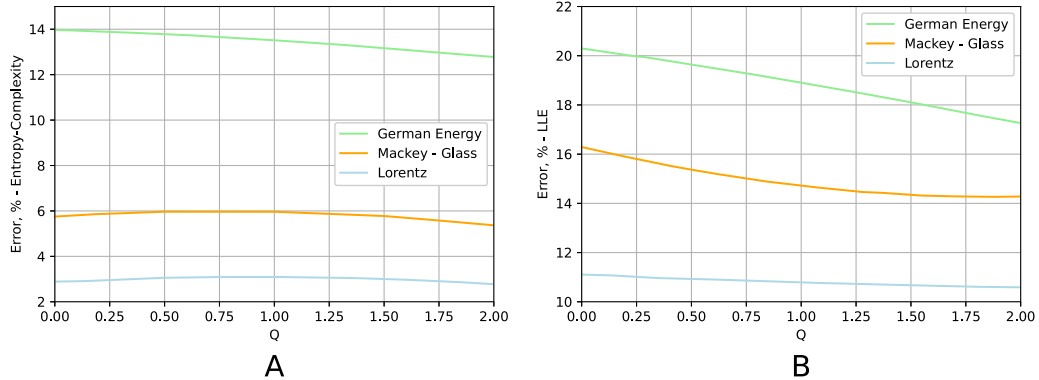

**Figure A6** A relative error for the position on the entropy-complexity plane (A) and for the largest Lyapunov exponent (B) for different values of the parameter **Q**. $N = 60$, $P_{max} = 2$, $S = 3$, $\Delta p = p_0$, $\Delta p_a = 10^{-6}\Delta p$.

## Competing Interests

The authors declare there are no competing interests.

## Author Contributions

- Vasilii A. Gromov conceived and designed the experiments, authored or reviewed drafts of the article, and approved the final draft.
- Yury N. Beschastnov performed the experiments, analyzed the data, performed the computation work, prepared figures and/or tables, and approved the final draft.
- Korney K. Tomashchuk performed the experiments, analyzed the data, performed the computation work, prepared figures and/or tables, authored or reviewed drafts of the article, and approved the final draft.

## Data Availability

The electricity load by country dataset and the scripts are available in the Supplemental Files.

The second dataset (Lorenz series) is not provided as it can be generated from the system of differential equations. The process is described at the beginning of the Results.

## Supplemental Information

Supplemental information for this article can be found online at http://dx.doi.org/10.7717/peerj-cs.1254#supplemental-information.

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
