# Peer review of "Generalized relational tensors for chaotic time series"

_PeerJ Computer Science, doi:10.7717/peerj-cs.1254_

## Round 0.1 · original submission · Major Revisions

Dear Authors,

Your manuscript has been reviewed. The reviewers have found your work interesting, but it needs significant revisions before being accepted for publication on PeerJ Computer Science.

In particular:

1) The terminology used in your paper has to be standardized to those adopted in the literature. Furthermore, the advancement in the field has to be taken into account, if any;
2) The description of the proposed approach should be more precise and detailed;
3) The algorithms shown in the paper should be presented using i) a pseudocode to formally introduce them and ii) a description of the rationale for each introduced procedure.

Reviewer 1 ·

Basic reporting

The manuscript is aimed at presenting a novel approach for representing and generating data using time series. Unfortunately, poor reporting significantly limits my understanding of the approach and, therefore, my evaluation of the manuscript’s merit. The manuscript should be thoroughly rewritten.
The major issues I faced while reading the manuscript.

I1. English. The language should be improved.

I2. Terminology. The authors should use accepted terms. The literature is vast. There are reviews, so the consistency between the used terminology and those used in the literature is crucial. For example, the manuscript’s first term (line 20), “discrete structure”, misleads the reader and makes the whole Introduction looks disconnected from the previous research. As a result, the statement (lines 69-73) that “none of the available discrete structures makes it possible to re-generate a series using the filled structure only” is at least controversial. In the majority of cited papers, the generation was included in the consideration. Thus, if the “discrete structure” corresponds to a “network”, then the statement (lines 69-73) is incorrect, and, as a result, there is no manuscript subject. Note that “series” can not be used to denote “time series”. Also, the authors call the approach a “tensor”. However, I do not see the corresponding mathematical structure. I disagree that tending to zero hourly electricity load values in Germany can be modelled by a low-dimensional chaotic system.

I3. The manuscript could be focused. The literature review forms comments but should provide a ground for understanding the presented approach. What are other approaches close to the considered one? Since there are reviews, the classification of approaches is available. The authors could start by putting their approach to the existing hierarchy. In such a way, the authors could identify competitors and possible bechmarking (comparison). In which papers a similar task was considered, and how? What were the outcomes?

Experimental design

I4. The description of the approach should be more precise and detailed. Is it a mathematical or computational approach? All notations should be explained. I do not see how Python could understand the instruction “If the position selected is already filled”. Note that the code should include comments according to usual practice. The description should be such that the algorithm could be reproduced by a reader.

Validity of the findings

I5. The purpose of the section Results remains a mystery to me. The description is scattered and introduces new terminology which was not discussed in the approach’s description. The statement that Figure 5 shows a typical trajectory of the Lorenz system has no ground.

Additional comments

In general, the authors should decide what the manuscript's purpose is. Is it to state that there are some Python scripts which can be used by experience people? Or is it the approach and codes which can be understandable and usable for other people?

·

Basic reporting

The authors introduce a generalized relational tensor to store (compress) the necessary information of a time series. This tensor, if I understand correctly, is like a multidimensional histogram that counts the number of observations that one would encounter at a certain time point and within a certain interval (i) of the observed value. After collecting enough time series samples, the tensor represents an empirical probability density function that can be used to reconstruct and predict the original time series. In particular, if S=2, then the relation tensor is nothing but the transition matrix of a Markov chain (but I suspect it is more than a transition matrix, since the tensor contains not just conditional probability distributions but a full multidimensional distribution, and thus can be used not only for discriminative purposes but for generative purposes as well.)

The paper is well structured. The results appear to be sound and have strong practical merits. Therefore, I recommend its publication. However, I have a few questions that should be addressed:

1. The use of tensor structures to improve the expressivity of predictive models for times series is not new, especially in the field of machine learning:
a. Yang, Y., Krompass, D. & Tresp, V. Tensor-Train Recurrent Neural Networks for Video Classification. in Proceedings of the 34th International Conference on Machine Learning (eds. Precup, D. & Teh, Y. W.) 70, 3891–3900 (PMLR, 2017).
b. Schlag, I. & Schmidhuber, J. Learning to Reason with Third Order Tensor Products. in Proceedings of Neural Information Processing Systems 2018 (eds. Bengio, S., Wallach, H., Larochelle, H., Grauman, K., Cesa-Bianchi, N. & Garnett, R.) 31, 9981–9993 (Curran Associates, Inc., 2018).
c. Yu, R., Zheng, S., Anandkumar, A. & Yue, Y. Long-term Forecasting using Higher Order Tensor RNNs. arXiv:1711.00073v3 (2019).
d. Meng, X. & Yang, T. Entanglement-Structured LSTM Boosts Chaotic Time Series Forecasting. Entropy 23, 1491 (2021).
which should be reflected in the paper. (However, the use of tensorization in these ML models lacks interpretability, which is the strength of this work.)

2. I do not understand Eq. (1). The left-hand side, lambda, is a “discrete structure” (what is a discrete structure, by the way?); but the right-hand side represents a norm of errors (so a number?). They do not match.

3. What are the dimensions of the generalized relational tensor? The authors should write them down explicitly and give a more intuitive (instead of mathematical) interpretation of the tensor.

4. Overused/misused language. For example, Line 53-54: “...please, for example, refer to (Mutua et al. (2016), Laut and Rath (2016))…”; line 56: “(please, refer to Takens’s theorem)”; line 80, “Maybe…”; line 366: “…The scalability of the algorithm in question is seems to be of fundamental importance…”

5. Line 167: \Lambda^* should be L^*.

6. All figures are blurry, and the label fonts are sometimes too small. The authors should use vector-based or higher-resolution images.

7. The authors should number the two Definitions (Lines 221 and 226).

8. “non-successive” -> “non-consecutive”.

Experimental design

The research question is well defined. The methods description contains sufficient details.

Validity of the findings

no comment

---

## Round 0.2 · Minor Revisions

Dear Authors
The paper needs minor revisions before it is accepted for publication in PeerJ Computer Science. More precisely:

1) The manuscript terminology must be revised following the reviewers' suggestions;
2) The introduction section must be rewritten following all the reviewers' indications.

·

Basic reporting

The authors have properly addressed my previous comments.

My only hesitation is that in the modified Introduction section, the authors claim that “…the only correct way to assess the quality of the representation of a time series is the similarity of the characteristics of the original and restored time series. Such characteristics can be Lyapunov spectrum, dimensions of attractors, entropy-complexity, partial autocorrelation function, etc. If this requirement is not met, then any judgments about the time series, made based on the structure with which the time series is represented, are groundless…”

In my opinion, this claim is dubious and misleading. Despite the very strong claim of “(being) the only correct way,” the claim was followed by very ambiguous words such as “similarity” and, even worse, “characteristics.” What do these words mean exactly? The authors give some examples to explain the “characteristics” they have in mind, but they do not provide criteria on what “characteristics” exactly should be included. Do the maximum/minimum, or even just the length of the time series account as “characteristics” as well? The authors must be very careful before rushing to make such a strong claim.

“...At the same time, we believe that only the information encoded in the structure should be used to restore the series...” Again, this is an ambiguous claim made by the authors. By definition, the information encoded in the discrete structure is ALL the information that is available. What else information can one use?

I suggest the authors think carefully again their choice of terminology. Here are some suggestions:

“discrete structures/representations”: I believe that the authors actually mean “discrete time-series models,” right?

“similarity”: this should be replaced by something like “quantitative difference between Lyapunov spectrum/attractor dimensions/etc of the original time series and the synthetic time series generated by the discrete model.”

The Introduction must be carefully rewritten before accepting the paper for publication.

Experimental design

no comment

Validity of the findings

no comment

Additional comments

no comment

---

## Round 0.3 · accepted · Accept

According to the reviewer's comments, the paper can be accepted for publication in its present form.

·

Basic reporting

The authors have satisfactorily addressed all my comments. I therefore recommend publication.

Experimental design

n/a

Validity of the findings

n/a